# A systematic review of population-based studies on lipid profiles in Latin America and the Caribbean

Rodrigo M Carrillo-Larco[1,2,3]*, C Joel Benites-Moya[2], Cecilia Anza-Ramirez[2], Leonardo Albitres-Flores[2,4,5], Diana Sánchez-Velazco[2], Niels Pacheco-Barrios[2], Antonio Bernabe-Ortiz[2,6]

[1]Department of Epidemiology and Biostatistics, School of Public Health, Imperial College London, London, United Kingdom; [2]CRONICAS Centre of Excellence in Chronic Diseases, Universidad Peruana Cayetano Heredia, Lima, Peru; [3]Universidad Católica Los Ángeles de Chimbote, Instituto de Investigación, Chimbote, Peru; [4]Facultad de Medicina de la Universidad Nacional de Trujillo, Trujillo, Peru; [5]Sociedad Científica de Estudiantes de Medicina de la Universidad Nacional de Trujillo-SOCEMUNT, Trujillo, Peru; [6]Universidad Científica del Sur, Lima, Peru

**Abstract** We aimed to study time trends and levels of mean total cholesterol and lipid fractions, and dyslipidaemias prevalence in Latin America and the Caribbean (LAC). Systematic-review and meta-analysis of population-based studies in which lipid (total cholesterol [TC; 86 studies; 168,553 people], HDL-Cholesterol [HDL-C; 84 studies; 121,282 people], LDL-Cholesterol [LDL-C; 61 studies; 86,854 people], and triglycerides [TG; 84 studies; 121,009 people]) levels and prevalences were laboratory-based. We used Scopus, LILACS, Embase, Medline and Global Health; studies were from 1964 to 2016. Pooled means and prevalences were estimated for lipid biomarkers from ≥2005. The pooled means (mg/dl) were 193 for TC, 120 for LDL-C, 47 for HDL-C, and 139 for TG; no strong trends. The pooled prevalence estimates were 21% for high TC, 20% for high LDL-C, 48% for low HDL-C, and 21% for high TG; no strong trends. These results may help strengthen programs for dyslipidaemias prevention/management in LAC.

*For correspondence: rcarrill@ic.ac.uk

Competing interests: The authors declare that no competing interests exist.

## Introduction

There is a growing body of evidence about levels, patterns and trends of body mass index, (*NCD Risk Factor Collaboration (NCD-RisC), 2019*; *NCD Risk Factor Collaboration (NCD-RisC), 2017a*) diabetes, (*NCD Risk Factor Collaboration (NCD-RisC), 2016*) blood pressure and hypertension, (*NCD Risk Factor Collaboration (NCD-RisC), 2017b*; *Geldsetzer et al., 2019*) yet much less has been reported about dyslipidaemias and cholesterol (*Farzadfar et al., 2011*; *NCD Risk Factor Collaboration (NCD-RisC), 2020a*). Unlike Latin America and the Caribbean (LAC), other world regions have multi-country studies or systematic reviews that have informed public health officers and practitioners about the burden of unhealthy lipid profiles (*Noubiap et al., 2018*). Moreover, available evidence already suggests there are non-trivial differences in lipid levels with other regions that deserve further scrutiny (*Farzadfar et al., 2011*; *Ponte-Negretti et al., 2017a*; *Ponte-Negretti et al., 2017b*). These facts show that regional evidence on lipid profiles and trends is limited in LAC, hampering the formulation of health policies and practice guidelines to prevent, treat and control dyslipidaemias with a regional focus.

This dearth of evidence has relevant implications for public health, clinical medicine and research in LAC. It is unknown whether surveillance systems are urgently needed to monitor dyslipidaemias, because the current cholesterol levels and whether they have increased or decreased have not been

**eLife digest** Cholesterol and triglycerides are fatty substances found in the blood. They are crucial components of cell membranes and important for a variety of processes in the body. But, too much, or too little blood fat can damage the blood vessels. For example, high levels of fat in the blood can clog arteries, which can increase the chances of heart disease, heart attacks and strokes.

Fat starts to build up if 'bad' fats, such as triglycerides and LDL cholesterol, are too high. But it can also happen if levels of 'good' fats, like HDL cholesterol, are too low. The causes of, and treatments for, these different types of dyslipidaemia (or fat levels outside normal ranges) are not the same. So, to plan interventions effectively, public health authorities need to know which type of blood fat imbalance is most common in the local population, and whether this has changed over time. In many parts of the world, this kind of information is available, but in Latin America and the Caribbean the data is incomplete.

To address this, Carrillo-Larco et al. reviewed around 200 previous studies from across Latin America and the Caribbean. This revealed that, since 2005, low HDL cholesterol has been the most common type of dyslipidaemia in this region, followed by elevated triglycerides, and third, high LDL cholesterol. These patterns have changed little over the years.

In many parts of the world, public health guidelines for dyslipidaemia focus on treatment specifically for high LDL cholesterol. But this new data suggests that guidelines should also include recommendations for HDL cholesterol, in particular in Latin America and the Caribbean. And, with a clearer understanding of the current pattern of blood fat imbalances in this region, researchers now have a baseline against which to measure the success of any new health policies. In the future, a multi-country study to measure blood fats in the general population could provide even more detail. But, until then, this work provides a starting point for customised health interventions.

quantified in LAC. In this line, if public health authorities should secure access to lipid-lowering medications in the current struggle to extend universal health coverage, (*Atun et al., 2015*) is also unknown because we have not quantified, which dyslipidaemia is the most prevalent in LAC. Finally, research cannot efficiently advance if LAC does not know what evidence is already available; thereby, resources can be targeted to where information is scarce or null.

Therefore, we aimed to provide robust evidence about trends of mean levels of total cholesterol and lipid fractions, as well as trends of dyslipidaemias prevalence in LAC. This evidence will guide policies and interventions so that they can focus on the most pressing issues. Also, public health officers can use this information as a starting point for disease surveillance and to monitor progress of interventions or to track targets.

## Results

### Selection process

The search yielded 6699 titles and abstracts; of these, 1123 were studied in detail and finally 197 studies met the inclusion criteria (*Figure 1—figure supplement 1*).

### General characteristics of selected studies

Brazil, with 61 studies, and Chile with 21 studies, contributed with the greatest number of studies to the systematic review (*Figure 1 - Figure 1—figure supplement 1* and *Supplementary file 1*, table 1). There were more studies conducted since 2010 (*Figure 1—figure supplement 1*). Across studies, the mean proportion of men in the study population was 43%, and the mean age was 48 years (*Supplementary file 1*, table 1).

### Total cholesterol

Evidence from 86 studies (168,553 individuals) informed the overall estimates on mean total cholesterol. The random-effects meta-analysis revealed a pooled mean total cholesterol of 193 mg/dl since 2005 (*Table 1*). During the last years, there seemed to be a negative yet weak correlation with time, signalling a small decrease in mean total cholesterol (*Figure 2*). National studies tended to report

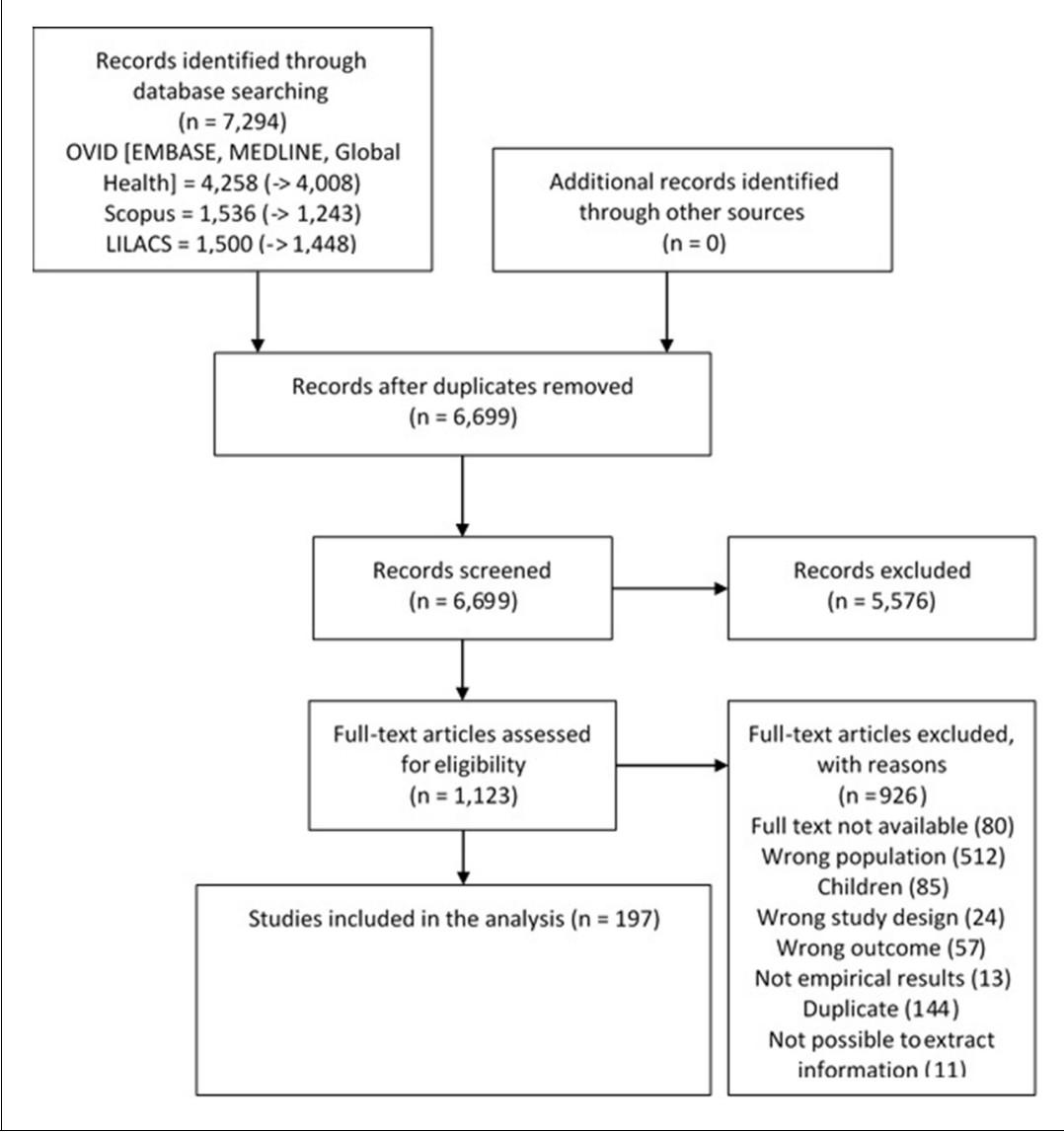

**Figure 1.** Study selection process.
The online version of this article includes the following figure supplement(s) for figure 1:

**Figure supplement 1.** Selection process and number of data sources per year.

lower mean levels than community and sub-national studies (*Figure 2*). Southern and Tropical Latin America appeared to have higher levels than the other sub-regions (*Figure 2*).

The total cholesterol prevalence estimates were informed by 68 studies (129,123 individuals) overall. The pooled prevalence since 2005 was 21% for total cholesterol ≥240 mg/dl and 34% for total cholesterol ≥200 mg/dl (*Table 1*). There was a positive trend with time, signalling an increase yet weak evidence supported this observation (*Figure 3*). National studies were evenly distributed; Southern and Tropical Latin America seemed to have higher estimates (*Figure 3*).

## LDL-Cholesterol

The overall sample for LDL-Cholesterol was 61 studies (86,854 subjects). Since 2005, the pooled mean was 120 mg/dl (*Table 1*). There was a non-significant decreasing trend (*Figure 2*). National studies seemed to report lower means, and there was not a clear geographic distribution (*Figure 2*).

Overall, LDL-Cholesterol prevalence estimates were informed by 29 studies (42,900 individuals). The pooled prevalence of high LDL-cholesterol since 2005 was 21% for LDL-Cholesterol ≥160 mg/dl

**Table 1.** Pooled mean and pooled prevalence estimates since 2005, random-effects meta-analysis.

|  | Mean (mg/dl) | Lower 95% confidence interval | Upper 95% confidence interval |
|---|---|---|---|
| Total cholesterol (37 studies) | 193.39 | 189.10 | 197.68 |
| LDL-Cholesterol (30 studies) | 119.98 | 116.08 | 123.88 |
| HDL-Cholesterol (42 studies) | 46.55 | 44.99 | 48.12 |
| Triglycerides (39 studies) | 139.27 | 130.57 | 147.98 |
|  | Prevalence (%) | Lower 95% confidence interval | Upper 95% confidence interval |
| High total cholesterol - ≥ 200 mg/dl (six studies) | 34.04 | 19.04 | 49.04 |
| High total cholesterol - ≥ 240 mg/dl (five studies) | 20.97 | 13.51 | 28.43 |
| High LDL-Cholesterol - ≥ 130 mg/dl (two studies) | 40.41 | 29.05 | 51.78 |
| High LDL-Cholesterol - ≥ 160 mg/dl (five studies) | 19.73 | 11.57 | 27.89 |
| Low HDL-Cholesterol ≤ 40(men) and ≤50(women) (nine studies) | 48.27 | 36.31 | 60.22 |
| High Triglycerides ≥ 150 mg/dl (12 studies) | 43.12 | 35.40 | 50.85 |
| High Triglycerides ≥ 200 mg/dl (four studies) | 20.48 | 16.28 | 24.69 |

and 40% for LDL-Cholesterol ≥130 mg/dl (*Table 1*), and such estimates have slightly increased

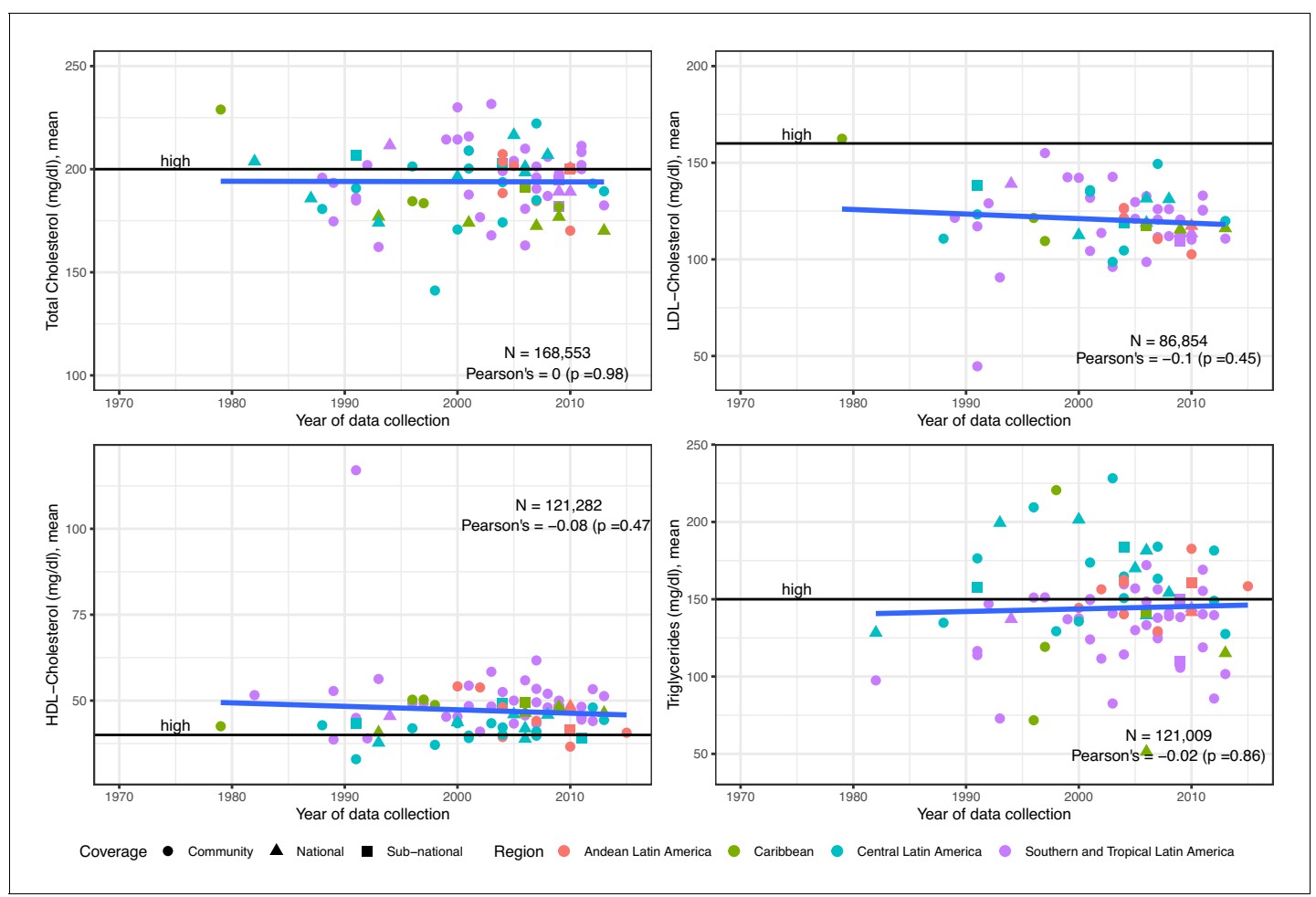

**Figure 2.** Trends in mean total cholesterol, LDL-Cholesterol, HDL-Cholesterol and triglycerides in Latin America and the Caribbean. The solid blue line represents a linear regression trend. Year in the x-axis refers to data collection year. Countries within sub-regions are shown in *Supplementary file 1*, Table 5. Individual estimates are shown in *Supplementary file 1*, Table 2.

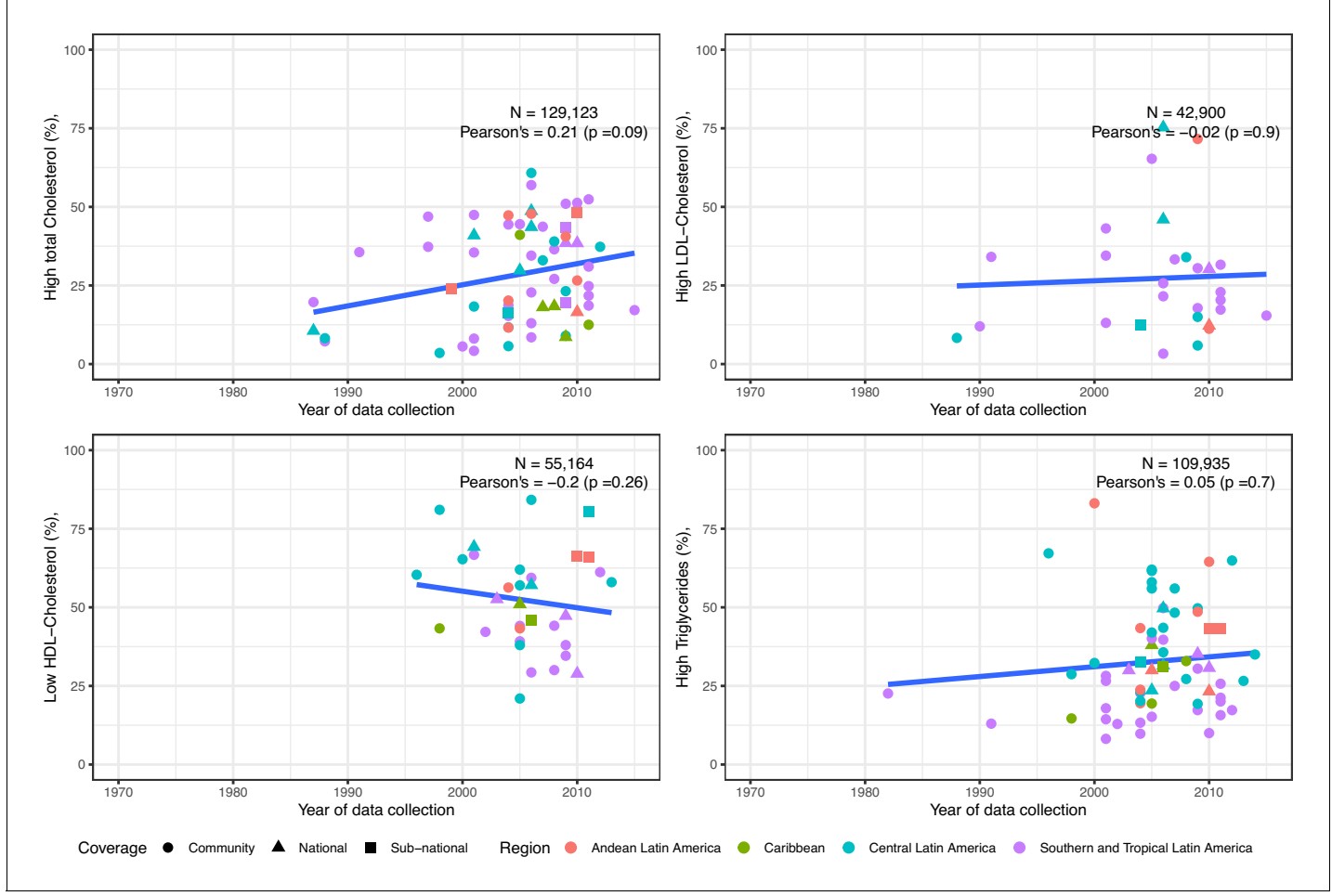

**Figure 3.** Trends in prevalence of high total cholesterol, high LDL-Cholesterol, low HDL-Cholesterol and high triglycerides in Latin America and the Caribbean. The solid blue line represents a linear regression trend. Year in the x-axis refers to data collection year. Countries within sub-regions are shown in ***Supplementary file 1***, Table 5. Studies included in this graphics are those with standard clinically relevant definitions. I.e., total cholesterol ≥150 mg/dl,≥200 mg/dl, or ≥240 mg/dl; LDL-Cholesterol ≥100 mg/dl,≥130 mg/dl, or ≥160 mg/dl; HDL-Cholesterol ≤40 mg/dl in men and ≤50 mg/dl in women; triglycerides ≥ 100 mg/dl,≥130 mg/dl,≥150 mg/dl, or ≥200 mg/dl. That is, prevalence estimates based on other definitions were excluded. Individual estimates are shown in ***Supplementary file 1***, Table 3.

(***Figure 3***). National studies were evenly distributed along the other studies (***Figure 3***). Southern and Tropical Latin America appeared to have higher estimates (***Figure 3***).

## HDL-Cholesterol

The HDL-Cholesterol mean estimates benefited from 84 studies (121,282 subjects). The pooled mean since 2005 was 47 mg/dl (***Table 1***). The time trend of mean HDL-Cholesterol was negative yet non-significant (***Figure 2***). National studies were evenly distributed; Southern and Tropical Latin America seemed to show higher means (***Figure 2***).

The HDL-Cholesterol prevalence estimates were based on 34 studies (55,164 individuals) overall. The pooled prevalence since 2005 was 48% for HDL-Cholesterol ≤40 mg/dl in men and ≤50 mg/dl in women (***Table 1***). The prevalence of low HDL-Cholesterol had a negative trend, yet not strong evidence supported this finding (***Figure 3***). National studies were evenly distributed; Central Latin America seemed to have higher rates of low HDL-Cholesterol (***Figure 3***).

## Triglycerides

There were 84 studies (121,009 people) included in the mean triglycerides analysis. The pooled mean was 139 mg/dl since 2005 (***Table 1***). The mean levels of triglycerides have slightly increased

(*Figure 2*). Estimates from national studies did not show a strong pattern (*Figure 2*). Estimates from Andean and Central Latin America appeared to be higher than those from Southern and Tropical Latin America (*Figure 2*).

Data from 70 studies (109,935 people) informed the triglycerides prevalence estimates overall. The pooled prevalence of high triglycerides was 21% for triglycerides $\geq$ 200 mg/dl and 43% for triglycerides $\geq$ 150 mg/dl (*Table 1*). The prevalence of high triglycerides has increased (*Figure 3*), yet weak evidence supported this finding. National studies did not show any patterns (*Figure 3*). As it was the case in mean triglycerides, Andean and Central Latin America seemed to have higher rates of high triglycerides (*Figure 3*).

## Risk of bias

Given the overall selection criteria (population-based studies with blood samples to measure lipid biomarkers), studies had moderate risk of bias. For details about the assessment tool and our grading rationale, please refer to *Supplementary file 1* pp. 08–09.

## Discussion

We summarized trends in total cholesterol, HDL-Cholesterol, LDL-Cholesterol and triglycerides; in addition, we also reported on trends of dyslipidaemias with clinical relevance: high total cholesterol, high LDL-Cholesterol, low HDL-Cholesterol and high triglycerides. This work, along with other global estimates, (*NCD Risk Factor Collaboration (NCD-RisC), 2020a*) can be used for surveillance of lipid levels in LAC. We have pooled recent mean and prevalence estimates, which can serve as a starting point to monitor changes during the next years. This work can also inform policies and interventions so that they can target the dyslipidaemia with the highest prevalence. Moreover, this work can inform regional practice guidelines to include local evidence and address regional needs.

Although there has been a marginal decrease in the mean of the four lipid biomarkers over time, the results do not support there have been a substantial change. A substantial change was not observed for prevalence estimates either. These findings are consistent with those from a recent global analysis, in which they found little change in total cholesterol and non-HDL Cholesterol in LAC, a decrease in several areas of North America, Europe, and Oceania, while an increase in East and South East Asia (*NCD Risk Factor Collaboration (NCD-RisC), 2020a*). These remarks may suggest that there have been few policies or interventions to improve these lipid profiles in LAC; alternatively, this could suggest that available interventions were not effective. In either case, the results may show a natural progression or variation, rather than the influence of any interventions to improve lipid profiles in LAC. Nonetheless, a potential explanation could be that effective policies were in place and these prevented an increase. A pending task in LAC is a comprehensive and quantitative evaluation of policies to decrease the burden of cardio-metabolic risk factors.

Overall, the global work by Taddei and colleagues suggested that lipid levels have decreased in several countries in North America, Europe and Oceania, yet lipid levels have increased in East and South East Asia; as we have discussed before, their findings for LAC agrees with ours showing little change over time (*NCD Risk Factor Collaboration (NCD-RisC), 2020a*). These patterns largely mirrors trends in cardiovascular disease mortality: death rates per 100,000 people in Eastern and Western Europe have decreased, death rates have increased in East and South Asia, while changes are modest in LAC (*IHME, GHDx, Viz Hub, 2020*). Lipid biomarkers are key cardio-metabolic risk factors, thus successful improvement in these at the patient and population level could bring gains in terms of cardiovascular outcomes reduction.

Hypercholesterolemia awareness and control may be low in LAC, as it has been exemplified in some cities (*Silva et al., 2010*; *Hernández-Alcaraz et al., 2020*; *Lotufo et al., 2016*); we do not have any strong evidence to assume this profile has improved since then. Because awareness, treatment and control for hypertension are still insufficient, (*Geldsetzer et al., 2019*) despite the fact that antihypertension drugs may have better availability than lipid-lowering medications, we hypothesize poor treatment rates for dyslipidaemias in LAC. Therefore, this potential poor awareness, treatment and control rates for dyslipidaemias may have translated into the unremarkable trends herein reported for LAC. A recent global analysis also found that lipid levels have changed little in LAC, while in many high-income countries these levels have improved (*NCD Risk Factor Collaboration*

(*NCD-RisC), 2020a*); this could be a consequence of better awareness and access to treatment in the latter countries.

In comparison to other world regions, a recent work located High−income English−speaking countries, Europe and High−income Asia−Pacific with larger mean levels of total cholesterol and HDL-Cholesterol than LAC (*NCD Risk Factor Collaboration (NCD-RisC), 2020a*). Diet and physical activity profiles, along with unequal access to primary prevention strategies, could be behind this difference. For non-HDL-Cholesterol, their findings revealed fewer regions above LAC, (*NCD Risk Factor Collaboration (NCD-RisC), 2020a*) which could indicate low prescription of lipid-lowering drugs (e.g., statins) in LAC. Time trends reported by the NCD-RisC largely agrees with our results, depicting LAC sub-regions with a marginal decrease since 1980 with regards to mean total cholesterol, HDL-cholesterol and non-HDL-Cholesterol (*NCD Risk Factor Collaboration (NCD-RisC), 2020a*). Other world regions have experienced a marked decrease or increase; this may support our hypothesis that little attention have been paid to lipid levels in LAC, resulting in unremarkable time changes.

The meta-analysis showed that the most frequent dyslipidaemia in LAC since 2005 was low HDL-Cholesterol. This could have relevant implications for clinical practice guidelines. Major international guidelines (*Grundy et al., 2019*) as well as guidelines from LAC (refer to *Supplementary file 1*, Table 4 for a summary of guidelines from selected countries in LAC) have solid algorithms and recommendations on when to start pharmacological treatment. Nonetheless, statins would reduce LDL-Cholesterol with little impact on HDL-Cholesterol. Consequently, based on our results, it would be advisable to strengthen clinical guidelines with further evidence, recommendations and algorithms to improve HDL-Cholesterol; these should include strong primary prevention strategies (e.g., lifestyles modification). This does not imply that lowering LDL-Cholesterol should not be important in LAC; on the other hand, this suggests that raising HDL-Cholesterol should also be addressed by guidelines and practitioners.

Our findings suggest that low HDL Cholesterol is the most common dyslipidaemia trait in LAC since 2005. Reasons behind this finding can relate to other cardio-metabolic risk factors. While raised body mass index (e.g., obesity) and diabetes have a negative correlation with HDL-Cholesterol, exercise has a positive effect on this lipid fraction (*Rashid and Genest, 2007*). The proportion of the population with obesity and diabetes has raised substantially throughout LAC, (*NCD Risk Factor Collaboration (NCD-RisC), 2019*; *NCD Risk Factor Collaboration (NCD-RisC), 2017a*; *NCD Risk Factor Collaboration (NCD-RisC), 2016*; *NCD Risk Factor Collaboration (NCD-RisC), 2020b*) which also happens to be the region with one of the largest frequencies of physical inactivity (*Guthold et al., 2018*). Even if clinical guidelines incorporate thorough recommendations to improve HDL-Cholesterol, this may not be achieved without policies or population-based interventions addressing the underlying (concomitant) cardio-metabolic risk factors. To successfully increase HDL-Cholesterol in LAC, reducing obesity and diabetes, while providing opportunities to do physical activity, are also needed.

This is a comprehensive systematic review conducted in five major search engines. However, limitations should be acknowledged. First, we did not search in grey literature sources; although in these sources we could have found some reports based on national surveys (e.g., WHO STEPS results), we argue that the results are sufficiently robust to have been largely influenced or driven by research published in grey literature. Second, although we selected only population-based studies to report on the scenario at the general population level, we did not apply other criteria to avoid potential health or clinical bias. For example, we did include a report if it had excluded people on lipid lowering medications. These studies could provide slightly higher estimates, than if they had included both people with and without lipid-lowering drugs. To the best of our knowledge, there has not been a systematic assessment of lipid-lowering medication uptake across LAC, yet we argue that these drugs are still not widely prescribed. Because there is evidence suggesting limitations in accessing hypertension and diabetes medication, (*Attaei et al., 2017*; *Chow et al., 2018*) we believe that these restrictions would be even greater for lipid-lowering drugs. In this context, we argue that this limitation may have had little impact on our estimates. Third, there were some years for which we did not retrieve any data. This limitation would not have affected the most recent trends and meta-analysis. In this line, we also did not present results for every country in LAC, limiting the extrapolation of our findings to these nations. Future studies with other analytical approaches could improve these limitations and provide information for all years and countries, or

even for sub-regions within LAC (*Farzadfar et al., 2011*; *NCD Risk Factor Collaboration (NCD-RisC), 2020a*). Fourth, although we only studied adult populations, the age range of the study participants may have not been the same across all selected studies. This could have biased our estimates if, the mean levels or prevalence estimates of the herein studied lipid biomarkers were significantly larger in some age ranges. Extracting published data to verify this hypothesis would considerably reduce the number of observations because not all studies reported their findings by (consistent) age groups. This would also be the case for sex, as studies would not always stratify findings by sex. Fifth, extraction of other characteristics of the studies could have provided relevant information to interpret the results and to draw the scenario of lipid research in LAC; among others, relevant features could include whether point-of-care devices were used or blood samples were analysed in laboratories, and whether these followed an international standard. Sixth, LDL-cholesterol could be measured directly or estimated (e.g., Friedewald formula), though this information was not extracted to further characterize the results. The Friedewald formula is frequently used, and it may underestimate the real value (*Meeusen et al., 2014*). If so, our estimates for LDL-cholesterol need to be interpreted cautiously; further research is needed to understand the magnitude of this potential underestimation and, if needed, to develop a more accurate formula for populations in LAC. In this line, a recent global work reported slightly larger mean levels of non-HDL Cholesterol in comparison to our LDL-cholesterol levels (*NCD Risk Factor Collaboration (NCD-RisC), 2020a*). Although these are not identical metrics, the agreement between these is typically good. Speculatively, we could hypothesize that some of surveys herein summarized used the Friedewald formula, and the LDL-cholesterol levels were underestimated. This could potentially explain the different results (*NCD Risk Factor Collaboration (NCD-RisC), 2020a*).

Some studies only reported the mean or the prevalence estimate. Although the prevalence estimates are relevant, from a public health perspective the mean and the shape of the distribution are also important and should be reported whenever possible. This calls for authors and reporting guidelines to provide both metrics. Similarly, studies did not report their results by sex or (consistent) age groups, precluding us to make estimates by gender and age.

A global endeavour reported on levels of total cholesterol for each country in the world until 2010 (*Farzadfar et al., 2011*; these work has been recently updated (*NCD Risk Factor Collaboration (NCD-RisC), 2020a*). The evidence from LAC was limited in comparison to our work, which also expands the evidence to include prevalence estimates. Beyond the CARMELA study which comprised seven cities in LAC, (*Pramparo et al., 2011*) and the CESCAS study which included cities in three countries, (*Rubinstein et al., 2011*) there is a dearth of multi-country studies addressing lipid profiles and other cardio-metabolic risk factors in LAC. These studies and other local research suggested that low HDL-Cholesterol was the most common dyslipidaemia. Our work confirms this observation and strengthens the evidence so that it can inform policies, interventions and guidelines.

Recently, an international work updated the 2010 global total cholesterol estimates (*Farzadfar et al., 2011*) and provided results for HDL-Cholesterol and non-HDL-Cholesterol (*NCD Risk Factor Collaboration (NCD-RisC), 2020a*). Our work complements this evidence by providing mean levels for other lipid fractions and prevalence estimates of clinically relevant dyslipidaemias in LAC. Their total cholesterol mean estimates for LAC are largely consistent with our findings, and so are the mean levels for HDL-Cholesterol; however, their non-HDL-Cholesterol means are larger than our LDL-cholesterol means. The reasons could be different methodology and analytical approach (please, refer to the limitations section).

Our findings, as those by the NCD-RisC, (*NCD Risk Factor Collaboration (NCD-RisC), 2020a*) suggested that mean levels of lipid biomarkers are not the same across LAC countries. Although LAC hosts mostly middle-income countries, there are large within countries inequalities; this is also seen in the Caribbean, where some islands may be high-income countries, but inequalities still exist. Different levels of poverty, access to healthy foods, opportunities for physical activity, and a still fragile primary health system, may explain the differences between sub-regions and countries in LAC.

A seminal work in Africa followed a similar methodology and reported, for the general population, a prevalence of 23% for high total cholesterol (our estimates were seven percentage points higher); 41% for low HDL-Cholesterol (our estimates were seven percentage points higher); 25% for elevated LDL-Cholesterol (our estimates were 15 percentage points higher); for triglycerides their prevalence was 16% (our results were 23 percentage points higher) (*Noubiap et al., 2018*). They

also reported estimates for other populations (e.g., people with diabetes) (*Noubiap et al., 2018*). The higher -worse- estimates we reported for LAC could suggest LAC is ahead in the epidemiological and nutritional transition, in comparison with Africa.

An interesting finding, which deserves in-depth scrutiny, is that we found a marginal decrease in mean total cholesterol, yet also a marginal increase in the prevalence of high total cholesterol. We propose two hypotheses. First, aging of the population. Older people may have larger prevalence of dyslipidaemias. As populations are aging and living longer, we could expect larger prevalence estimates, while mean levels get 'diluted' or do not necessarily change. Second, and closely related, is that we do not know the drivers of these changes, i.e., we still need to answer whether the mean or the tails of the distribution are changing and driving the trends. For blood pressure, it has been suggested that the mean is the main driver of trends in raised blood pressure prevalence (*NCD Risk Factor Collaboration (NCD-RisC), 2018*); whether this is the case for lipid biomarkers and dyslipidaemias is still unknown. A third option could be the uptake of lipids-lowering drugs. As more people take these drugs, the population mean would decrease while the prevalence would not change (or even increase). However, as we have argued before, lipids-lowering medication coverage may still be limited in LAC.

Improving prevention, care and management for diabetes and hypertension is a clear priority globally and in LAC (*González-Villalpando et al., 1999*; *Hall Martínez et al., 2005*; *Hernández-Hernández et al., 2017*). Nonetheless, lipid profiles are relevant for public health and clinical practice as well (*Grundy et al., 2019*; *NICE, 2014*). In fact, they are a major risk factor for cardiovascular events including ischemic heart disease and stroke (*Lewington et al., 2007*; *Di Angelantonio et al., 2009*). Also, lipid biomarkers are predictors in several cardiovascular risk scores used to guide treatment allocation for primary cardiovascular prevention (*Goff et al., 2014*; *WHO CVD Risk Chart Working Group, 2019*; *Hajifathalian et al., 2015*). This work provides timely regional evidence to start a research and policy agenda to improve lipid profiles in LAC.

Our work has compiled the largest number of data sources across years and countries in LAC. Consequently, it is uniquely positioned to inform local and regional authorities about recent trends in lipids distribution and prevalence estimates. The results could have multiple pragmatic applications. First, they could be used as a baseline upon which build surveillance systems to monitor future trends of lipid biomarkers. Second, our comprehensive search strongly suggests that there is a lack of evidence from several countries, particularly in Central America and the Caribbean. Local and regional authorities should conduct epidemiological studies or population-wide surveys (e.g. STEPS approach); alternatively, when these have already been conducted, data could be open access for research purposes. Ideally, where data are available, these should meet the FAIR acronym: findable, accessible, interoperable, and reusable. Unfortunately, the first two elements of the acronym are perhaps the least frequent, yet the most important for scientific use of available data. Third, our results could inform prevention strategies and policies. Because we have reported on different lipid fractions, medication (e.g., statins) could be prioritized where LDL-cholesterol is higher, while diet or healthy lifestyles could be a priority where HDL seems to be the most important issue.

Levels and prevalence estimates of unhealthy total cholesterol, LDL-Cholesterol, HDL-Cholesterol and triglycerides seemed not to have substantially changed over the last years in LAC. Since 2005 across LAC, the most common dyslipidaemia was low HDL-Cholesterol. These results should inform policies so that they can start or strengthen strategies to improve lipid profiles, thus reducing the burden of cardiovascular events which are strongly associated with unhealthy lipid levels.

## Materials and methods

### Protocol
This is a systematic review of the literature (PROSPERO CRD42019120491; PRISMA Checklist available in *Supplementary file 1*). We aimed to identify trends in total cholesterol and cholesterol fractions in LAC general population; also, to ascertain which dyslipidaemia (e.g. low HDL-cholesterol) is the most prevalent in LAC.

## Eligibility criteria

Research reports were analysed if they targeted adult men and women of the general population. We focused on LAC populations, thus studies with LAC populations in countries outside the LAC region, and studies with only foreign populations in LAC nations, were excluded. Population-based studies were defined as those which followed a random sampling of the general population. Conversely, studies addressing specific populations (e.g., shanty towns), those with patients (e.g., stroke survivors), or people with risk factors (e.g., smokers), were excluded.

The outcomes of interest were lipid biomarkers levels and dyslipidaemia prevalence. We focused on clinically and public health relevant lipid biomarkers: total cholesterol, HDL-Cholesterol, LDL-cholesterol and triglycerides. Only studies in which lipid biomarkers were measured with valid methods (e.g. laboratory or point-of-care devices) were included; that is, studies which results relied only on self-reported information were excluded.

## Information sources

The search was conducted on December 21$^{st}$, 2018. We used Scopus, LILACS, Embase, Medline and Global Health; the last three through Ovid. In all of these, the search was conducted without time or language restriction. The search terms are available in *Supplementary file 1* pp. 06–07.

## Study selection

Results from each search engine were downloaded and saved in EndNote where duplicates were dropped. A second search for duplicates was conducted with Rayyan, an online tool for systematic reviews (*Ouzzani et al., 2016*). Titles and abstracts were independently reviewed by two researchers (RMC-L and NP-B; CA-R and CJB-M; LA-F and DS-V), and discrepancies were solved by consensus or a third party (AB-O). After this screening phase, selected reports were downloaded and independently studied in detail by two researchers (RMC-L and NP-B; CA-R and CJB-M; LA-F and DS-V); discrepancies were solved by consensus or by a third party (AB-O). Finally, selected studies were scrutinized again to check for data duplication, i.e. different reports that used the same data (e.g., a national survey). In this case, the paper which presented more information (e.g., all four lipid biomarkers), or the one with the largest sample size, was included in the systematic review and meta-analysis. In other words, we aimed to include each study or survey once. The unit of analysis is a study.

## Data collation

An extraction form was developed by the authors and tested with a random sample of selected studies; the form was not modified after data collation started. This form included study's characteristics: mean age, proportion of men, year of data collection, and if it was a nationally representative sample. The extraction form also collated the mean and prevalence estimate as well as a dispersion measurement (e.g. standard deviation or confidence interval) of the available lipid biomarkers.

## Risk of bias of individual studies

We used the risk of bias tool developed by Hoy and colleagues (*Hoy et al., 2012*). Notably, this tool was also used by a systematic review on a similar topic (*Noubiap et al., 2018*). These criteria were implemented in an Excel spreadsheet and evaluated independently by two reviewers (RMC-L and NP-B; CA-R and CJB-M; LA-F and DS-V); discrepancies were solved by consensus or a third party (AB-O).

## Summary measures

We present both a narrative and quantitative summary. The narrative summary described the study's characteristics, while the quantitative summary explored the trends of the lipid biomarkers means as well as prevalence estimates. In addition, following a random-effects meta-analysis and using data from 2005 onwards, we computed the pooled mean and the pooled prevalence estimate for each lipid biomarker and dyslipidaemia trait. We only used the most recent data (i.e., from 2005) to report on the current -or most recent- levels in LAC, rather than summarizing all available information with no clear time frame. Because the selected studies had different sample size and scope (e.g., national surveys versus community studies), we conducted the random-effects meta-analysis. Unlike a fixed-

effect meta-analysis, in a random-effects meta-analysis large studies would not drive or bias the pooled estimates.

## Ethical considerations

This is a systematic review of published scientific evidence and open information. Human subjects did not participate in this work directly and there was no intervention. Approval from an IRB/ethics committee was not requested.

## Role of the funder

The funder had no role in the research question, data collation, analysis or reporting of the results. All the authors collectively are responsible for data accuracy and they all have approved the submitted work.

---

# Additional information

### Funding

| Funder | Grant reference number | Author |
| --- | --- | --- |
| Wellcome Trust | 214185/Z/18/Z | Rodrigo M Carrillo-Larco |

The funders had no role in study design, data collection and interpretation, or the decision to submit the work for publication.

---

### Author contributions

Rodrigo M Carrillo-Larco, Conceptualization, Formal analysis, Investigation, Writing - original draft, Writing - review and editing; C Joel Benites-Moya, Cecilia Anza-Ramirez, Leonardo Albitres-Flores, Diana Sánchez-Velazco, Niels Pacheco-Barrios, Investigation, Writing - review and editing; Antonio Bernabe-Ortiz, Conceptualization, Investigation, Writing - review and editing

### Author ORCIDs

Rodrigo M Carrillo-Larco https://orcid.org/0000-0002-2090-1856
Cecilia Anza-Ramirez https://orcid.org/0000-0001-7364-8252
Leonardo Albitres-Flores http://orcid.org/0000-0002-0077-3615
Antonio Bernabe-Ortiz http://orcid.org/0000-0002-6834-1376

### Decision letter and Author response

Decision letter https://doi.org/10.7554/eLife.57980.sa1
Author response https://doi.org/10.7554/eLife.57980.sa2

---

# Additional files

### Supplementary files

• Supplementary file 1. Word document with checklist for reporting of systematic reviews; search terms used; risk of bias assessment and rationale; table describing the selected studies along with references; table with the mean levels by selected study; table with prevalence estimates by selected study.

• Transparent reporting form

### Data availability

This is a systematic review and meta-analysis of scientific literature. Data used came from scientific manuscripts which can be accessed online through libraries and scientific repositories. All relevant information is included within the manuscript.

---

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
