## [Decision Letter]

**Acceptance summary:**

While cholesterol levels have been decreasing in Europe, North America and Australasia and increasing in many parts of Asia with corresponding changes in cardiovascular mortality cholesterol levels in South America and the Carribean have changed very little. The authors comprehensively review the available data from South America and the Carribean and point out the need for action.

**Decision letter after peer review:**

Thank you for submitting your article "A systematic review of population-based studies on Lipid profiles in Latin America and the Caribbean" for consideration by *eLife*. Your article has been reviewed by three peer reviewers, including Edward D Janus as the Reviewing Editor and Reviewer #1, and the evaluation has been overseen by Matthias Barton as the Senior Editor. The following individuals involved in review of your submission have agreed to reveal their identity: Brian Tomlinson (Reviewer #2); Ian Hambleton (Reviewer #3).

The reviewers have discussed the reviews with one another and the Reviewing Editor has drafted this decision to help you prepare a revised submission.

You will see that we see this as a very good piece of work and have gone over it in considerable detail and have raised these points to assist you in providing clarity. We appreciate that you may not be able to readily address them all.

Please use this title: "A systematic review of population-based studies on Lipid profiles in Latin America and the Caribbean".

Summary:

The authors have conducted a very thorough systematic review of the historical and current evidence on lipid profiles in Latin America and the Caribbean. It has included a very large number of studies and subjects and it provides an important synthesis of the available literature, from a region that is a little under-represented in the peer-reviewed literature, and that sometimes has evidence in less accessible journals. The analysis has been performed appropriately and the manuscript is well written. It is worthy of publication, once several substantive and minor comments have been adequately addressed. A really nice piece of work!

Essential revisions:

1) The Abstract could be improved by indicating the number of studies/subjects reviewed and the time period of the studies involved.

2) Subsection “Eligibility criteria**”**.

"Adult men and women included". How were studies with differential age ranges incorporated into the meta-analysis? And see points (14) and (16) below.

3) Search strategy.

The search strategy is very nicely laid out in the Supplementary file. The absence of grey literature for national survey work (Discussion section) may be more influential than the authors imagine. Many national surveys from the Caribbean (for example) are not published in peer reviewed literature. This is not a deal-breaker – it is reasonable to not include a grey literature search, but this missing block of (steps) surveys should be recognised. This also links to where the authors note that more surveys are needed (Discussion section). Might be good to refer to the fair data principles. Often, data are available, but not easily findable or accessible.

National vs regional data and policy could be mentioned.

4) The number of subjects in each study is not given in Supplementary file 1—table 1 nor are dot sizes in Figure 1 showing this. Also, it is unclear if Figure 2 and Figure 3 show the year of the survey or of the publication – these may well differ. You have this in the supplementary table but not in the legends for Figure 2 and Figure 3. You should also if possible, have a column for Point of care/Laboratory/Accredited reference laboratory and show this for each study in your supplementary table. It would be useful to comment if any of the studies compared subjects living in rural and urban areas.

5) Males and females are not reported separately Usually they differ especially for HDL-cholesterol and triglycerides. If possible, please provide this.

6) Methodology for LDL-cholesterol is not noted. Was it the Friedewald calculation or direct measurement? Please show in your Supplementary Table if possible.

7) For TG prevalence is provided from 127,718 observations but it is stated that there were only 77,435 individuals used to calculate the means. Why is this? Similarly, the numbers used to derive means and prevalence are unclear for HDL-cholesterol and LDL-cholesterol.

8) In pooling for means did this give appropriate weight to the study size e.g. the average when combining a small (A) and a large (B) study is not the average of the mean from study A and study B but closer to the mean of study B. This is not clear in the manuscript. Similar issues arise with prevalence. This needs to be very clear for the reader.

9) HDL-cholesterol is decreased by both overweight/obesity and by the presence of type 2 diabetes which have increased in populations worldwide. This should come into the discussion. Alcohol increases HDL and exercise also increases it so they should also be mentioned.

Given that you main interesting finding is low HDL and you think there should be intervention/ management of this discussion is critical.

10) It would be useful to provide further comment on how these lipid levels compare with other parts of the world. Although they are not ideal, they are probably not as bad as many other countries. You could also discuss briefly trends over time in Europe, Australia, New Zealand and North America where lipid levels have decreased – markedly in some countries e.g. Finland – since the 1960s while in Asia there have increases over the same time frame e.g. Japan, Korea. Your lack of data from the period 1960-1980 is a limitation in this respect and this can be noted as such.

11) There is some comment about different lipid levels in different countries or different regions. It may be interesting to expand that if possible.

12) Unit of Analysis.

– Usually, in this type of systematic review, the unit-of-analysis is the survey. We would (commonly) link multiple articles to a single survey and draw on all available evidence to build a picture of each survey. On a number of occasions in this report, the authors flip between talking about "studies" and about "articles" – and this leads to lack of clarity on the unit-of-analysis used. More is needed in the Materials and methods section on this. E.g. – the authors seem to select a single article as representative of a survey. But I could not work out which article represented each survey.

– And in subsection “Selection process” – "281 papers (305 studies) met the inclusion criteria". Again, this confuses me a bit with respect to unit-of-analysis. I would usually expect to see more articles than surveys. Perhaps we are saying that some articles reported on multiple surveys? We need a bit of clarity on this – some extra text, perhaps offered in the Supplement.

– The Supplementary file 1—table 1 seems to list multiple articles from a single survey (Barbados is a good example – 2 articles from the same survey). I would strongly advise including a subsection – "Unit of Analysis" – and clearing up any uncertainty about this throughout the text. And it would be great (in the Supplementary file 1—table 1) to link papers to studies. That would be a really useful part of creating more clarity around the unit-of-analysis.

– Subsection “General characteristics of selected reports” is another example. Brazil (e.g.) has 66 data sources. But the real importance here is not the number of articles found, but the number of studies that contribute to the analysis from Brazil…

– And subsection “Total cholesterol”. "158,947 individuals informed the estimates". We should be able to read the number of studies as well – remembering to *always* report the unit-of-analysis metrics. Same in subsection “LDL-Cholesterol”, etc.

13) Risk of Bias (RoB).

– RoB for observational work remains a developing methodological area – so it is tricky! Nonetheless, I would like to hear some more from the authors on this. The tool is not easy to find (a reference in a reference) – so could be included as a supplement. In discussing, the authors should consider the common dilemma between rating scales, and the more qualitative assessment favoured (e.g.) by the Cochrane Collaboration these days. Recognise as well, the importance of assessing study-quality rather than assessing article quality. And sometimes (regularly in fact) an article does not give us enough information to make a RoB determination.

– In subsection “Risk of Bias”. Risk of bias was deemed to be low for all studies. This jumps out as being overly optimistic. Recognising that I could not access the Risk of Bias tool used. It would be very unusual (indeed) for a large systematic review of observational evidence to find only low-bias surveys. Considerations such as confounder adjustment, level of missing data and so on should be playing an important role in bias linked to observational evidence. And regularly, RoB cannot be ascertained due to lack of article information.

14) Meta-analysis vs Meta-regression.

This meta-analysis I think includes at least one covariate (e.g. time) and perhaps others (e.g. sub-region?). I would suggest that the analysis might be better termed a "random-effects meta-regression". A more complete description would be useful in the Materials and methods section.

15) Data synthesis presentation.

The scatterplot visuals are fine – but do not allow a reader to identify individual studies. I strongly suggest that the authors include a Forest plot for each outcome. Can be in Supplement, and (crucially for a systematic review) allows the reader to also interpret work at the unit-of-analysis level – i.e. for each survey.

16) Summary-level vs individual participant data (IPD) analyses.

In subsection “General characteristics”, the authors note "Men in study population was 26%. Men age was 47 years". I have assumed that the meta-regression was at the summary-data level (in other words, study-level means, or prevalence rates were the regression inputs). If I have that right, how did the authors cope with studies having different age-range inclusions and studies that may or may not have used statistical weighting / adjustments to present their summary data? The only other method would be an IPD analysis – but I don't think the authors had access to individual-level survey data? I could be wrong – I can find no mention of IPD?

17) Discussion section – surveillance approach.

The authors note "This work used a surveillance approach". It’s not too clear to me what this means, within a systematic review context?

18) Discussion around the finding of little lipid change over time.

Authors conclude that either (A) few policies/interventions exist to reduce lipids or (B) interventions were ineffective. There is a third possibilities. That the policies and interventions have been pretty good, and without them the lipid profiles would have increased.

19) Discussion section – guidelines.

There were no guidelines from Caribbean listed in this supplement. If none, this might warrant mention? But… I believe there are national and regional Caribbean examples. E.g. www.paho.org/hq/index.php?option=com_content&view=article&id=1423:2009-managing-hypertension-primary-care-caribbean&Itemid=1353&lang=en

HEARTS international movement also can be mentioned (World Health Organization).

[Editors' note: further revisions were suggested prior to acceptance, as described below.]

Thank you for resubmitting your work entitled "A systematic review of population-based studies on Lipid profiles in Latin America and the Caribbean" for further consideration by *eLife*. Your revised article has been evaluated by a Reviewing Editor and a Senior Editor.

We thank you for the comprehensive responses and revision of the manuscript which adequately address the reviewers comments. While the manuscript has been much improved there is still one issue that need to be addressed before it can be accepted for publication.

Please take note of the recent publication by Taddei et al.et al., (2020) The study by Taddei et al.et al., includes analyses on Latin America and the Caribbean and thus should be cited and discussed in the context of the findings presented in your revised manuscript.

---

## [Author Response]

Summary:The authors have conducted a very thorough systematic review of the historical and current evidence on lipid profiles in Latin America and the Caribbean. It has included a very large number of studies and subjects and it provides an important synthesis of the available literature, from a region that is a little under-represented in the peer-reviewed literature, and that sometimes has evidence in less accessible journals. The analysis has been performed appropriately and the manuscript is well written. It is worthy of publication, once several substantive and minor comments have been adequately addressed. A really nice piece of work!

We appreciate the positive feedback and fully agree with the statement that Latin America and the Caribbean (LAC) is “a little under-represented in the peer-reviewed literature”. We are focusing our work to improve this situation. Working with editors and reviewers like you, makes this task gratifying. Thank you very much indeed.

Essential revisions:1) The Abstract could be improved by indicating the number of studies/subjects reviewed and the time period of the studies involved.

We have included the analysed number of studies and the underlying sample size for each lipid fraction; we have also included the time period. To avoid taking too much space in the abstract, the sample size has been reported along with the abbreviations. The modified Abstract reads: “… lipid (total cholesterol [TC; 86 studies with 168,553 people], HDL-Cholesterol [HDL-C; 84 studies with 121,282 people], LDL-Cholesterol [LDL-C; 61 studies with 86,854 people], and triglycerides [TG; 84 studies with 121,009 people]) levels and prevalences were laboratory-based. We used Scopus, LILACS, Embase, Medline and Global Health; summarized studies were from 1964 to 2016.”

2) Subsection “Eligibility criteria**”**."Adult men and women included". How were studies with differential age ranges incorporated into the meta-analysis? And see points (14) and (16) below.

This is very interesting caveat, which we did not account for. We have further discussed this issue in the Discussion section. “Fourth, although we only studied adult populations, the age range of the study participants may have not been the same across all selected studies. […] Extracting published data to verify this hypothesis would considerably reduce the number of observations, because not all studies reported their findings by (consistent) age groups.”

3) Search strategy.The search strategy is very nicely laid out in the Supplementary file 1. The absence of grey literature for national survey work (Discussion section) may be more influential than the authors imagine. Many national surveys from the Caribbean (for example) are not published in peer reviewed literature. This is not a deal-breaker – it is reasonable to not include a grey literature search, but this missing block of (steps) surveys should be recognised. This also links to where the authors note that more surveys are needed (Discussion section). Might be good to refer to the fair data principles. Often, data are available, but not easily findable or accessible.National vs regional data and policy could be mentioned.

We believe the reviewers made a remarkable point about STEPS surveys and grey literature. We have toned down the Discussion section; the new text reads: “First, we did not search in grey literature sources; although in these sources we could have found some reports based on national surveys (e.g., WHO STEPS results), we argue that the results are sufficiently robust to have been largely influenced or driven by research published in grey literature.”

We have further discussed about the FAIR acronym; this read: “Ideally, where data are available, these should meet the FAIR acronym: findable, accessible, interoperable, and reusable. Unfortunately, the first two elements of the acronym are perhaps the least frequent, yet the most important for scientific use of available data.”

4) The number of subjects in each study is not given in Supplementary file 1—table 1 nor are dot sizes in Figure 1 showing this. Also, it is unclear if Figure 2 and Figure 3 show the year of the survey or of the publication – these may well differ. You have this in the supplementary table but not in the legends for Figure 2 and Figure 3. You should also if possible, have a column for Point of care/Laboratory/Accredited reference laboratory and show this for each study in your supplementary table. It would be useful to comment if any of the studies compared subjects living in rural and urban areas.

The sample size has been included in the Supplementary file. Also, in the footnotes for Figure 2 and Figure 3 we have specified: Year in the x-axis refers to data collection year. The figures have been updated to also include this information in the x-axis label.

Unfortunately, we did not extract any information on whether POC devices were used, or whether samples were processed in laboratories. We did not collate any information on rural/urban location either; most studies did not report their results stratified by age, sex or urban/rural location. We have discussed this potential limitation: “Fifth, extraction of other characteristics of the original studies could have provided relevant information to interpret the results and to draw the scenario of lipid research in LAC; among others, relevant features could include whether point-of-care devices were used or blood samples were analysed in laboratories, and whether these followed an international standard.”

5) Males and females are not reported separately Usually they differ especially for HDL-cholesterol and triglycerides. If possible, please provide this.

We agree with the fact that potential sex difference matters. However, we did not extract this information. As we discussed about age groups, most reports did not have results by sex. This has been discussed: “This would also be the case for sex, as studies would not always stratify findings by sex.”

6) Methodology for LDL-Cholesterol is not noted. Was it the Friedewald calculation or direct measurement? Please show in your Supplementary Table if possible.

Unfortunately, this information was not extracted, though discussed in the Discussion section: “Sixth, LDL-cholesterol could be measured directly or estimated (e.g., Friedewald formula), though this information was not extracted to further characterize the results. […] This could potentially explain the different results.”

7) For TG prevalence is provided from 127,718 observations but it is stated that there were only 77,435 individuals used to calculate the means. Why is this? Similarly, the numbers used to derive means and prevalence are unclear for HDL-cholesterol and LDL-cholesterol.

This is because we did not conduct an individual-level meta-analysis; in other words, we extracted summary information from published reports. These reports were more likely to present prevalence estimates rather than numeric results (e.g., means). This is the reason why the sample size for means and prevalences are not the same. We had discussed this: “Some studies only reported the mean or the prevalence estimate. Although the prevalence estimates are relevant, from a public health perspective the mean and the shape of the distribution are also important and should be reported whenever possible…”

8) In pooling for means did this give appropriate weight to the study size e.g. the average when combining a small (A) and a large (B) study is not the average of the mean from study A and study B but closer to the mean of study B. This is not clear in the manuscript. Similar issues arise with prevalence. This needs to be very clear for the reader.

Not particularly, however, this was one of the reasons why we did a random-effects meta-analysis (rather than fixed-effect). In a random-effects meta-analysis, we expected that each study will contribute to the pooled estimates proportional to its sample size, without biasing the results towards those studies with larger influence. We have complemented this information in the subsection “Summary measures”: “Because the selected studies had different sample size and scope (e.g., national surveys versus community studies), we conducted the random-effects meta-analysis. Unlike a fixed-effect meta-analysis, in a random-effects meta-analysis large studies would not drive or bias the pooled estimates.”

9) HDL-cholesterol is decreased by both overweight/obesity and by the presence of type 2 diabetes which have increased in populations world wide. This should come into the discussion. Alcohol increases HDL and exercise also increases it so they should also be mentioned.Given that you main interesting finding is low HDL and you think there should be intervention/ management of this discussion is critical.

We have incorporated a new paragraph in the Discussion section to address these topics: Our findings suggest that low HDL-cholesterol is the most common dyslipidaemia trait in LAC since 2005. Reasons behind this finding can relate to other cardio-metabolic risk factors. While raised body mass index (e.g., obesity) and diabetes have a negative correlation with HDL-cholesterol, exercise has a positive effect on this lipid fraction (Rashid and Genest, 2007). The proportion of the population with obesity and diabetes has raised substantially throughout LAC (NCD Risk Factor Collaboration (NCD-RisC), 2019; NCD Risk Factor Collaboration (NCD-RisC), 2017; NCD Risk Factor Collaboration (NCD-RisC), 2016; NCD Risk Factor Collaboration (NCD-RisC), 2020), which also happens to be the region with one of the largest frequencies of physical inactivity. (Guthold et al., 2018). Even if clinical guidelines incorporate thorough recommendations to improve HDL-cholesterol, this may not be achieved without policies or population-based interventions addressing the underlying (concomitant) cardio-metabolic risk factors. To successfully increase HDL-cholesterol in LAC, reducing obesity and diabetes, while providing opportunities to do physical activity, are also needed.

10) It would be useful to provide further comment on how these lipid levels compare with other parts of the world. Although they are not ideal, they are probably not as bad as many other countries. You could also discuss briefly trends over time in Europe, Australia, New Zealand and North America where lipid levels have decreased – markedly in some countries e.g. Finland – since the 1960s while in Asia there have increases over the same time frame e.g. Japan, Korea. Your lack of data from the period 1960-1980 is a limitation in this respect and this can be noted as such.

We have incorporated two paragraphs in the Discussion section.

“Recently, an international work updated the 2010 global total cholesterol estimates (Farzadfar et al., 2011) and provided results for HDL-cholesterol and non-HDL-cholesterol (NCD Risk Factor Collaboration (NCD-RisC), 2020). […] The reasons could be different methodology and analytical approach (please, refer to the limitations section).”

“In comparison to other world regions, a recent work located High−income English−speaking countries, Europe and High-income Asia-Pacific with larger mean levels of total cholesterol and HDL-cholesterol than LAC. […] Other world regions have experienced a marked decrease; this may support our hypothesis that little attention have been paid to lipid levels in LAC, resulting in unremarkable time changes.”

11) There is some comment about different lipid levels in different countries or different regions. It may be interesting to expand that if possible.

These comments were in the Results section. We have further elaborated on these in the Discussion section: “Our findings, as those by the NCD-RisC (NCD Risk Factor Collaboration (NCD-RisC), 2020), suggested that mean levels of lipid biomarkers are not the same across LAC countries. […] Different levels of poverty, access to healthy foods, opportunities for physical activity, and a still fragile primary health system, may explain the differences between sub-regions and countries in LAC.”

12) Usually, in this type of systematic review, the unit-of-analysis is the survey… We would (commonly) link multiple articles to a single survey and draw on all available evidence to build a picture of each survey. On a number of occasions in this report, the authors flip between talking about "studies" and about "articles" – and this leads to lack of clarity on the unit-of-analysis used. More is needed in the Materials and methods section on this. E.g. – the authors seem to select a single article as representative of a survey. But I could not work out which article represented each survey.

The unit of analysis is a study. When multiple reports analysed the same study, then we chose the report (or paper) with the most information or the largest sample size. Throughout the manuscript we have changed the terms to “study”. Also, we have included the following in the Materials and methods section: “That is, we aimed to include each study or survey once. The unit of analysis is a study.”

– And in subsection “Selection process” – "281 papers (305 studies) met the inclusion criteria". Again, this confuses me a bit with respect to unit-of-analysis. I would usually expect to see more articles than surveys. Perhaps we are saying that some articles reported on multiple surveys? We need a bit of clarity on this – some extra text, perhaps offered in the Supplement.

Apologies for the lack of clarity. As explained above, we have changed the term to “study” throughout the manuscript. Figure 1—figure supplement 1 has been updated as well.

– The Supplementary file 1—table 1 seems to list multiple articles from a single survey (Barbados is a good example – 2 articles from the same survey). I would strongly advise including a subsection "Unit of Analysis" – and clearing up any uncertainty about this throughout the text. And it would be great (in the Supplementary file 1) to link papers to studies. That would be a really useful part of creating more clarity around the unit-of-analysis.

This was an error we have corrected. As we explained in the methods and above (please refer to point 12, we aimed to include each survey (e.g., STEPS) or study just once. We have verified again all studies included and eliminated duplicate surveys/studies. The total sample size of studies included in this work is now 197. The manuscript has been updated accordingly without significant changes in the results, interpretation and conclusions.

– Subsection “General characteristics of selected reports” is another example. Brazil (e.g.) has 66 data sources. But the real importance here is not the number of articles found, but the number of studies that contribute to the analysis from Brazil…

We have changed as suggested, to show the number of studies these countries contributed to the review. This read: “*Brazil, with 61 studies, and Chile with 21 studies, contributed with the greatest number of studies to the systematic review…”*

– And subsection “Total cholesterol”. "158,947 individuals informed the estimates". We should be able to read the number of studies as well – remembering to *always* report the unit-of-analysis metrics. Same in subsection “LDL-Cholesterol”, etc.

In these lines, we have included the number studies too. If the reviewers and editors allow, we would like to keep the sample size (along with the number of studies as requested), these has been reported in parenthesis next to the number of studies.

13) Risk of Bias (RoB).– RoB for observational work remains a developing methodological area – so it is tricky! Nonetheless, I would like to hear some more from the authors on this. The tool is not easy to find (a reference in a reference) – so could be included as a supplement. In discussing, the authors should consider the common dilemma between rating scales, and the more qualitative assessment favoured (e.g.) by the Cochrane Collaboration these days. Recognise as well the importance of assessing study-quality rather than assessing article quality. And sometimes (regularly in fact) an article does not give us enough information to make a RoB determination.

We included the reference to the RoB tool (as well as a reference to another similar review), but probably this was not clear enough. Thus, we have reorganized that paragraph to make the RoB tool reference much more evident: “We used the risk of bias tool developed by Hoy and colleagues (Hoy et al., 2012). Notably, this tool was also used by a systematic review on a similar topic (Noubiap et al., 2018).” Please, see the next answer for further explanations and arguments regarding the other proposed issues.

– In subsection “Risk of Bias”. Risk of bias was deemed to be low for all studies. This jumps out as being overly optimistic. Recognising that I could not access the Risk of Bias tool used. It would be very unusual (indeed) for a large systematic review of observational evidence to find only low-bias surveys. Considerations such as confounder adjustment, level of missing data and so on should be playing an important role in bias linked to observational evidence. And regularly, RoB cannot be ascertained due to lack of article information.

We are happy to change our overall opinion to “moderate risk of bias”; in fact, we have changed Results section to reflect this.

Below, we explain our rationale for each of the ten items of the RoB assessment tool we followed. Also, we elaborated on our overall final judgment. Finally, we discussed that the evaluation of one report by study, in contrast to, for example, the original protocol, may be a limitation. A similar version of the following lines has been included in the Supplementary file and referred where relevant in the main text.

“The first item in the risk of bias (RoB) tool is: Was the study’s target population a close representation of the national population in relation to relevant variables, e.g. age, sex, occupation? We considered this “low” risk of bias because we only included population-based studies with random sampling of the general population, and we excluded studies with one population group alone (e.g., smokers). Whether a community study is a close representation of the country, is of course arguable. However, that study would still be “more representative” than one with a selected or convenience sample. For this item the possible outcomes were “low risk of bias” and “high risk of bias”. We considered our selected studies as “low risk of bias” because even when they were not full national surveys, they may still be representative of the general population. […]

A limitation of our methodology is that we assessed RoB based on the information available in each selected report (or paper), which may not contain all details to make a comprehensive assessment of the original study. Ideally, we would have needed to investigate the original protocol, but certainly this does not happen often in any systematic review. Moreover, because of our original selection criteria, we strongly consider that the selected reports do not provide biased information to affect our results.”

14) Meta-analysis vs Meta-regression.This meta-analysis I think includes at least one covariate (e.g. time) and perhaps others (e.g. sub-region?). I would suggest that the analysis might be better termed a "random-effects meta-regression". A more complete description would be useful in the Materials and methods section.

Apologies for the confusion. We titled our work as “Meta-Analysis” because of the results in Table 1, where we present the pooled mean and prevalence estimates (following a random-effects meta-analysis). We did not conduct any meta-regression. When the reviewers refer to time, I believe they may be wondering about the results in Figure 2 and Figure 3, where we plotted the means and prevalences by year of data collection. For these plots we did not conduct a meta-regression based on time, and neither did we include other covariates (e.g., sub-region). Conversely, Figure 2 and Figure 3 are as simple as a scatter plot of the mean and prevalence estimates on the y-axis, and year of the data collection on the x-axis. In addition to this graphical representation (Figure 2 and Figure 3), we calculated the correlation, for those most interested in a “strong” number rather than visual inspection of time trends.

In summary, Figure 2 and Figure 3 are a graphical representation of how means and prevalences have changed over time, with a correlation coefficient; that is, these do not represent a meta-regression analysis. We considered our work as a meta-analysis because of the pooled estimates in Table 1.

15) Data synthesis presentation.The scatterplot visuals are fine – but do not allow a reader to identify individual studies. I strongly suggest that the authors include a Forest plot for each outcome. Can be in Supplement, and (crucially for a systematic review) allows the reader to also interpret work at the unit-of-analysis level – i.e. for each survey.

We understand that the issue in hand is transparency and facilitating the readers to see the information at the unit of analysis level (the study). We have included in supplementary materials two tables showing this information, i.e., all estimates at the study level. These tables have been referenced in the footnotes of the main figures; these are Supplementary file 1—table 2 and Supplementary file 1—table 3. If the editors and reviewers allow, we would prefer not to present forest plots, which are basic figures in meta-analyses. We consider our work tells the story of lipid levels and prevalences, and for that we took advantage of an established methodology (systematic review and meta-analysis); however, this work is not a “standard” systematic review with meta-analysis.

16) Summary-level vs individual participant data (IPD) analyses.In subsection “General characteristics”, the authors note "Men in study population was 26%. Men age was 47 years". I have assumed that the meta-regression was at the summary-data level (in other words, study-level means, or prevalence rates were the regression inputs). If I have that right, how did the authors cope with studies having different age-range inclusions and studies that may or may not have used statistical weighting / adjustments to present their summary data? The only other method would be an IPD analysis – but I don't think the authors had access to individual-level survey data? I could be wrong – I can find no mention of IPD?

The summary estimates extracted from the original reports were indeed at the study/summary-level. The overall mean age and mean male proportion, was a summary of summaries; i.e., the mean of the extracted means. We did not conduct a meta-regression analysis. Summaries extracted from the original reports were plotted by year of data collection (Figure 2 and Figure 3). The meta-analysis refers to the pooled estimates presented in Table 1. Given our overall methodology, and because we did not anticipate conducting a meta-regression analysis, we did not account for the different age ranges in the original selected studies. We have included this in the Discussion section; please, refer to our answer to question number 2 for further details.

17) Discussion section – surveillance approach.The authors note "This work used a surveillance approach". It’s not too clear to me what this means, within a systematic review context?

We have further elaborated on this idea: “…We have pooled recent mean and prevalence estimates, which can serve as a starting point to monitor changes during the next years.”

18) Discussion around the finding of little lipid change over time.Authors conclude that either (A) few policies/interventions exist to reduce lipids or (B) interventions were ineffective. There is a third possibilities. That the policies and interventions have been pretty good, and without them the lipid profiles would have increased.

We have incorporated this suggestion. This now reads: “Nonetheless, a potential explanation could be that effective policies were in place and these prevented an increase. A pending task in LAC is a comprehensive and quantitative evaluation of policies to decrease the burden of cardio-metabolic risk factors.”

19) Discussion section – Guidelines.There were no guidelines from Caribbean listed in this supplement. If none, this might warrant mention? But… I believe there are national and regional Caribbean examples. E.g. www.paho.org/hq/index.php?option=com_content&view=article&id=1423:2009-managing-hypertension-primary-care-caribbean&Itemid=1353&lang=enHEARTS international movement also can be mentioned (World Health Organization).

As also acknowledged in the manuscript, we chose a few selected guidelines. We did not aim to summarize all guidelines, neither to search for the guidelines or equivalent documents in all countries and territories in LAC. We believe this is beyond the scope of this work, and those we used served as examples. If the reviewers and editors feel this is not correct, we are happy to remove these lines and the Supplementary table. This would not affect the overall spirit of the work.

[Editors' note: further revisions were suggested prior to acceptance, as described below.]

Please take note of the recent publication by Taddei et al., (2020) The study by Taddei.et al., includes analyses on Latin America and the Caribbean and thus should be cited and discussed in the context of the findings presented in your revised manuscript.

According to the Editors, the authors need to address specifically the fact that – while lipid levels in South America and the Caribbean have changed little over time – mean cholesterol levels in North America, Europe, Australia and New Zealand have decreased over time while in most of Asia – East and South East – and also in the Pacific they have increased with corresponding impact on cardiovascular disease prevalence, especially in younger and middle aged individuals. The authors should expand on this and discuss this issue in their own words in the revised manuscript.

In our previous response, i.e., in the version we submitted after incorporating the reviewers’ comments, we included and discussed the cited work (Taddei et al., 2020). The work by Taddei and colleagues was published while we were working on the revisions. However, in this version we further elaborated on the specific points raised by the Editors.

First, we used this reference to strengthen an argument we made in the second paragraph of the Discussion section: “Although there has been a marginal decrease in the mean of the four lipid biomarkers over time, the results do not support there have been a substantial change. A substantial change was not observed for prevalence estimates either. These findings are consistent with those from a recent global analysis, in which they found little change in total cholesterol and non-HDL-cholesterol in LAC, a decrease in several areas of North America, Europe, and Oceania, while an increase in East and South East Asia (NCD Risk Factor Collaboration (NCD-RisC), 2020).”

Immediately after this we have introduced a new paragraph to specifically address the Editors’ request: “Overall, the global work by Taddei and colleagues suggested that lipid levels have decreased in several countries in North America, Europe and Oceania, yet lipid levels have increased in East and South East Asia; as we have discussed before, their findings for LAC agrees with ours showing little change over time (NCD Risk Factor Collaboration (NCD-RisC), 2020). […] Lipid biomarkers are key cardio-metabolic risk factors, thus successful improvement in these at the patient and population level could bring gains in terms of cardiovascular outcomes reduction.”

Second, we also used this reference and the idea suggested by the Editors to complement the arguments in the third paragraph of the Discussion section (now fourth after the inclusion the paragraph described above): “Therefore, this potential poor awareness, treatment and control rates for dyslipidaemias may have translated into the unremarkable trends herein reported for LAC. A recent global analysis also found that lipid levels have changed little in LAC, while in many high-income countries these levels have improved (NCD Risk Factor Collaboration (NCD-RisC), 2020); this could be a consequence of better awareness and access to treatment in the latter countries.”